Comparison of instrument-assisted soft tissue mobilization and proprioceptive neuromuscular stretching on hamstring flexibility in patients with knee osteoarthritis

Anjum Narmeen 1
Sheikh Raheela Kanwal 2
Omer Aadil 3 4
Anwar Kinza 3
Khan Muhammad Manan Haider 5
Aftab Anam 6
Awan Waqar Ahmed waqar.ahmed@riphah.edu.pk 3 7
1 Physiotherapy Department, Pakistan Railway Hospital, , Rawalpindi , Pakistan
2 College of Applied Medical Sciences, University of Hail , Hail , Saudi Arabia
3 Riphah College of Rehabilitation and Allied Health Sciences, Riphah International University, Islamabad , Islamabad , Pakistan
4 School of Rehabilitation, Tehran University of Medical Sciences , Tehran , Iran
5 Department of Rehabilitation Sciences, Shifa Tamer e Millat University Islamabad , Islamabad , Pakistan
6 Faculty of Pharmacy & Allied Health Sciences, University of Sialkot , Sialkot , Pakistan
7 Health Education Research Foundation , Islamabad , Pakistan
Lavender Andrew
Electronic publication date: 2023 Dec 1
Publication date: 2023
Volume: 11
Electronic Location ID: e16506
Received 2023 Mar 20; Accepted 2023 Nov 1
Copyright: ©2023 Anjum et al.
Copyright year: 2023
Copyright holder: Anjum et al.
License: This is an open access article distributed under the terms of the Creative Commons Attribution License, which permits unrestricted use, distribution, reproduction and adaptation in any medium and for any purpose provided that it is properly attributed. For attribution, the original author(s), title, publication source (PeerJ) and either DOI or URL of the article must be cited.
License URL: https://creativecommons.org/licenses/by/4.0/

Keywords: Hamstring muscle, Knee Osteoartheritis, Pain, Manual therapy, Stretching exercises

Funding: The authors received no funding for this work.

==============================
Background

The association between hamstring tightness and knee osteoarthritis (KOA) is significant because tight hamstrings can put more strain on the knee joint, reduce its range of motion, and cause compensatory movements that worsen the KOA.

Objective

To compare the effects of instrument-assisted soft tissue mobilization (IASTM) and proprioceptive neuromuscular (PNF) on hamstring flexibility in patients with KOA.

Methods

Data for the randomized controlled trial (NCT05110326) was collected from n = 60 participants randomly divided into group A received IASTM and group B received PNF stretching. In group A, the therapist made 30 strokes gentle strokes with the tool from the origin to the insertion while holding the plane at a 45-degree angle over the treatment area. In group B, PNF stretching was done with three repetitions and 10 seconds rest between each, after isometric contraction of the hamstring muscle using approximately 50% of their maximum strength, holding it for 8 seconds, and then releasing it. A 30-minute session was given to each patient three times per week and was given for 6 weeks. Outcome measures were the visual analog scale (VAS) for pain intensity, the active knee extension test (AKET) for hamstring flexibility, and the Western Ontario and McMaster Universities Arthritis Index (WOMAC) for the health status of KOA patients.

Results

The study found a significant interaction (p < 0.001) between interventions and time across several measurements. After 6 weeks, both interventions resulted in significant improvements (p < 0.001) across all dependent variables, with group A (IASTM) showing more significant improvement in hamstring flexibility, pain reduction, and health status (p < 0.001) compared to group B (PNF).

Conclusions

Both the IASTM technique and PNF stretching resulted in increased hamstring flexibility, decreased pain, and enhanced general health. The IASTM technique, however, showed potential benefits over PNF stretching in terms of flexibility, pain relief, and public health enhancement. Physical therapists and manual therapists may prioritize the usage of the IASTM technique for patients who want to make significant changes in these areas.

Introduction

Knee osteoarthritis (KOA) is a painful musculoskeletal disease caused by degeneration and articular cartilage loss over time. KOA mostly affects the elderly population and commonly causes disability throughout the world (Mora, Przkora & Cruz-Almeida, 2018). About 250 million people or 4% of the world’s population suffer from osteoarthritis (Kohn, Sassoon, & Fernando, 2016). KOA affects 28.0% of the urban population and 25.0% of the rural population in Pakistan (Yunus et al., 2022).

Age, congenital and acquired deformity, trauma and female gender are all risk factors that have been linked to the development of KOA. The overweight population is more prone to develop osteoarthritis. Joint injury or overuse damages the articular cartilage leading to osteoarthritis (Antony et al., 2016). Persistent knee discomfort, stiffness in the morning, and functional impairments are the major symptoms for diagnosing KOA. Additionally, crepitus, joint mobility limitation, and bony enlargement are all useful indicators of KOA (Alshami, 2014; Heidari, 2011; Yunus et al., 2022).

Knee joint movements occur by the two primary muscles including the quadriceps and hamstrings and assist the smooth and accurate ambulatory growth factors in the knee joint (Cavanellas et al., 2018; Jyoti & Yadav, 2019; Onigbinde et al., 2013). The association between hamstring tightness and KOA has significance because tight hamstrings can put more strain on the knee joint, reduce its range of motion, and cause compensatory movements that make the condition worse (Li et al., 2021; Miyamoto, Hirata & Kanehisa, 2017). It has also been observed that decrease mean hamstring flexibility has been found in KOA (Onigbinde et al., 2013). This decreased flexibility may increase the compressive stress on the patello-femoral joint and lead to patellofemoral syndrome, which often contributes to osteoarthritis and can result in pain and limitations in physical functioning (Jyoti & Yadav, 2019; Mahant & Shukla, 2021; Mullaney & Fukunaga, 2016; Onigbinde et al., 2013; Sherazi et al., 2022; Walli, McCay & Tiu, 2023).

There are different conventional approaches for the management of knee osteoarthritis including medical, surgical, and rehabilitative strategies. Physiotherapy includes the application of modalities, exercise therapy, and different manual therapy technique (Lim & Al-Dadah, 2022). The Instrument-Assisted Soft Tissue Mobilization (IASTM) technique that is the most frequent soft tissue mobilization techniques (Cheatham, Baker & Kreiswirth, 2019; Cheatham et al., 2016). Modern types of these IASTM interventions comprise various stainless-steel instruments such as Graston, GuaSha, Ergon, etc. (Avramova, Tsvetkova-Gaberska & Sport, 2022; Cheatham et al., 2016). It involves the use of a tool that causes mechanical micro-traumatic injury to the treated area. As a result, an inflammatory reaction is elicited, speeding up the healing process and restoring the flexibility and integrity of the tissue and cartilage healing (Fousekis et al., 2020). The therapeutic effects of this approach appear to include decreasing tissue adhesion, enhancing the number of fibroblasts, and stimulating collagen synthesis (Kiran et al., 2022; Cheatham, Baker & Kreiswirth, 2019).

In those with knee osteoarthritis (KOA), proprioceptive neuromuscular facilitation (PNF) stretching might be an effective technique to improve hamstring flexibility. This stretching method combines isometric contractions with passive stretching to increase joint range of motion and muscle flexibility (Gao et al., 2023). Research has shown that PNF stretching is more effective than conventional stretching at relieving pain in KOA patients (Moyano et al., 2013). PNF recognized to be more effective than other stretching techniques in enhancing passive and active flexibility. It may help maintain the balance of stress between the medial and lateral compartments at the knee (Lim, 2018; Sajedi, Bayram & Bilgiç, 2020; Shen et al., 2022).

Enhancing hamstring flexibility through exercise and stretching can aid in symptom relief and lessen pressure on the knee joint, making it a crucial component of managing and treating KOA. IASTM and PNF stretching are two therapies that have been utilized to improve muscle flexibility and range of motion in joints. However, it is unclear how well they perform compared to one another when it comes to enhancing hamstring flexibility in knee osteoarthritis patients. It was hypothesized that IASTM techniques are more effective than PNF stretching in improving hamstring flexibility among KOA patients. Therefore, the study aimed to compare the effectiveness of IASTM and PNF stretching on hamstring flexibility in patients with KOA.

Methods

Study design & setting

It was a single-blinded, randomized clinical trial (NCT05110326) conducted at the RHS Rehabilitation Centre (RHS/EC/28-06-2021-03), Islamabad, Pakistan. The study was completed within 1 year from July 2021-June 2022 and approval was taken from the research and ethical committee (REC) of the Faculty of Rehabilitation and Allied Health Sciences (with Ref# Riphah/RCRS/REC-01055) Riphah International University.

Participants

The 35–50 years patient having grade 1&2 knee osteoarthritis (KOA), according to the Kellegren and Lawrence criteria, hamstring tightness of more than 20° from the active knee extension test (AKET) (Kanishka et al., 2019), were included, through non-probability purposive sampling technique. Patients having lower extremity injury/surgeries in the past 6 months, any hip, or knee fractures or deformity, neurological symptoms, tightened iliotibial band, adductor muscle, and sartorius associated with other musculoskeletal conditions were excluded from the study.

Sample size

A total of n = 60 sample size was calculated through G power, keeping the effect size small (0.24), α error margin at 0.05. To avoid β error probability, the power (1−β) was set at 0.95%. A total of n = 63 patients were assessed for eligibility and n = 60 participants fulfilled the inclusion criteria and were randomly allocated to group A (n = 30), which received IASTM technique, and group B (n = 30) which received PNF technique. A total of n = 57 participants were analyzed at the end of study due to loss of follow-up of n = 3 patients from Group A (Fig. 1).

Figure 1 CONSORT diagram.

Randomization

The sealed enveloped method using a computerized random number generator was used for randomization. An individual who was not directly involved in the study did the random allocation. The random numbers were then written on the index cards and placed in a thick and opaque sealed envelope before the start of the study. After obtaining written informed consent, the physical therapist opened the envelope and provided the respective interventions to the patients. As the assessing physical therapist was blinded to the intervention so the study was single-blinded.

Intervention

A 30-minute session was given to each patient 3 times per week and was given for 6 weeks. Baseline assessments were made prior to beginning the intervention and re-assessment was done after 6 weeks. The detail intervention protocol for group A (Ganesh, Ayman & Allen, 2017; Kim & Lee, 2018) and group B (Meena, Shanthi & Madhavi, 2016) can be followed in Table 1.

Table 1 Intervention protocol.

	Group A (IASMT Technique)	Group B (PNF Stretching)	
	A hot pack was applied to the hamstring muscle and knee joint for 15 min before the IASTM and PNF technique.	
Position	•Prone position with knees bent at 45-degree angle	•Supine position with hips flexed to 90-degree angle	
Technique	•IASTM technique using contoured stainless-steel Ergon® instrument.
• Vaseline is applied to reduce friction between tool and skin.
• Tool cleaned with alcohol pad before use.
• 30 gentle strokes from origin to insertion at 45-degree angle	•Flex the knee against resistance provided by the therapist’s hand.
• Extend the knee until a slight stretching sensation is felt in the hamstring muscle.
• Make an isometric contraction of the hamstring muscle using approximately 50% of maximum strength.
• Hold the isometric contraction for 8 s.
• Release the isometric contraction.
• After mild to moderate stretching with no pain, therapist stretched the hamstring muscle further.
• Hold the stretch for 30 s.
• Perform PNF stretching with 3 repetitions.
• Allow 10 s of rest between each repetition.	
	The patient then received isometric quadriceps exercises, where they were instructed to press and hold onto a towel roll placed below the knee for 20 s. They performed 5 sets of exercises, each consisting of 10 repetitions.
At the end the patient then performed five sets of ten repetitions each of hamstring isometric exercises. The patient held a towel roll placed beneath the ankle in place for 20 s throughout each repeat.	

Assessments

The researchers considered the ethical, legal and regulatory norms and standards for this research according to the Declaration of Helsinki as a statement of ethical principles for medical research involving human subjects, including research on identifiable human material and data.

The visual analog scale (VAS) was used to determine the severity of pain. The patient marked a point on a 100-mm horizontal line that indicated a continuum between “no pain” and “worst discomfort” (Alghadir et al., 2018; Klimek et al., 2017). The VAS is the most reliable in measuring OA knee pain, with the smallest errors and excellent test–retest reliability (ICC = 0.97) (Alghadir et al., 2018).

The hamstring flexibility was assessed by active knee extension test (AKET) using a goniometer (Kim & Lee, 2018; Rahman et al., 2022). The AKET was shown to have high intra-rater agreement (ICC =.86–.99) and moderate inter-rater agreement (ICC =.76-.89) reliability (Hansberger et al., 2019).

The Western Ontario and McMaster Universities Arthritis Index (WOMAC) index was also used to assess the health status of individuals with knee osteoarthritis including three subscales of pain, stiffness, and physical function. It consisted of 24 questions with a total score ranging from 0–96 (Copsey et al., 2019; Hafez et al., 2013). All WOMAC subscales (pain, stiffness, and physical function) have satisfactory test-retest reliability with ICCs of 0.86, 0.68, and 0.89 (Nakarmi, Haq & Vaidya, 2019; Salaffi et al., 2003).

Statistical methods

The descriptive statistics, such as frequency (n), percentage (%), mean, standard deviation (SD), and mean differences (MD), used to summarize the study’s findings and subsequently presented in tables and graphs. A two-way mixed ANOVA with partial eta squared (ηp2) as the effect size was used to examine the interaction between interventions and the level of assessment because parametric tests were determined to be appropriate for the data, the paired-sample t-tests were used to analyze changes over time for within-group comparisons, while independent t-tests used for between-group comparisons. As a measure for effect size, Cohen’s d was employed. The data examined using SPSS version 28 with a significance level of p < 0.05.

Results

The mean age and BMI of the study participants were 45.14 ± 4.67 years and 28.53 ± 5.65 kg/m2, respectively. A total of n = 14 (24.6%) of the n = 57 participants were men, and the remaining n = 43(75.4%) were women. The n = 30 participants were found to have grade 1 knee osteoarthritis (KOA) according to the Kellegren and Lawrence criteria, whereas the remaining n = 27 were determined to have grade 2 KOA. Figures 2 and 3, respectively, show how the groups’ BMI and KOA grading distributions were distributed.

Figure 2 Frequency distribution (BMI).

Figure 3 Frequency distribution knee OA grades.

Figure 4 Interaction effect.

To find the interaction effect between intervention and level of assessment, two-way mixed ANOVA was applied. As the sphericity was assumed, the results showed that there was a significant interaction effect between both interventions and time factor in all dependent variables with large effect size, including hamstring flexibility on AKET {F =32.13(1,55), p < 0.001, ηp2 = 0.36}, pain on VAS {F =52.95(1,55), p < 0.001, ηp2 = 0.49} as well as pain {F =53.38(1,55), p < 0.001, ηp2 = 0.49}, stiffness {F =18.25(1,55), p < 0.001, ηp2 = 0.24}, physical function {F =78.26(1,55), p < 0.001, ηp2 = 0.58} and health status on WOMAC {F =96.15(1,55), p < 0.001, ηp2 = 0.63} as shown in Fig. 4

As there was a significant interaction effect for all the dependent variables, paired samples t-test was applied to determine the main effect. It showed that significant improvement (p < 0.001) was observed in all the dependent variables with large effect size including hamstring flexibility (AKET), pain (VAS), and health status (WOMAC) including pain, stiffness, and physical functions after the 6th week of intervention (Table 2).

Table 2 Baseline and after 6 weeks within group analysis.

		Group A (IASMT technique) n = 27	Group B (PNF stretching) n = 30	
		Mean	SD	MD	p-value	Cohen’s d	Mean	SD	MD	p-value	Cohen’s d	
AKE O	Baseline	41.01	3.26	−28.85	.000***	3.56	40.11	3.16	−22.38	.000***	4.86	
	After 6th week	69.87	1.92			62.50	3.82				
VAS (Pain)	Baseline	73.33	7.96	62.00	.000***	12.33	72.70	9.17	36.06	.000***	14.34	
	After 6th week	11.33	12.49			36.63	13.59				
WOMAC												
Pain	Baseline	10.37	2.11	5.92	.000***	1.68	9.46	3.25	2.56	.000***	1.77	
	After 6th week	4.44	1.45			6.90	2.75				
Stiffness	Baseline	3.77	1.36	2.37	.000***	1.11	4.53	1.56	1.26	.000***	0.82	
	After 6th week	1.40	0.88			3.26	1.36				
Physical function	Baseline	32.55	5.29	16.44	.000***	5.30	30.23	9.12	6.60	.000***	2.84	
	After 6th week	16.11	4.00			23.63	8.03				
Total Score	Baseline	46.62	6.76	24.48	.000***	6.71	44.23	12.04	10.43	.000***	3.74	
	After 6th week	22.14	5.74			33.80	10.58				
Notes.

AKE active knee extension test

VAS visual analogue scale

WOMAC Western Ontario and McMaster Universities Arthritis Index

Differences within groups were analyzed by independent sample t-test.

*** p < 0.001.

When comparing both groups, the results of the independent t-test showed that those who received IASTM technique, more significantly improved (p < 0.001) hamstring flexibility (AKET), pain (VAS), health status (WOMAC) with a large effect size than group B who received PNF technique (Table 3).

Table 3 Baseline and after 6 weeks between group analysis.

		Group A (IASTM technique) n = 27	Group B (PNF stretching) n = 30				
		Mean	SD	Mean	SD	MD	p-value	Cohen’s d	
AKET O	Baseline	41.01	3.26	40.11	3.16	−.90	0.29	–	
After 6th week	69.87	1.92	62.50	3.82	−7.37	0.00***	3.08	
VAS (Pain)	Baseline	73.33	7.96	72.70	9.17	−.63	.78	–	
After 6th week	11.33	12.49	36.63	13.59	25.30	0.00***	13.08	
WOMAC									
Pain	Baseline	10.37	2.11	9.46	3.25	−.90	0.22	–	
After 6th week	4.44	1.45	6.90	2.75	2.45	0.00***	2.23	
Stiffness	Baseline	3.77	1.36	4.53	1.56	.75	0.522	–	
After 6th week	1.40	0.88	3.26	1.36	1.85	0.00***	1.16	
Physical function	Baseline	32.55	5.29	30.23	9.12	−2.32	0.24	–	
After 6th week	16.11	4.00	23.63	8.03	7.52	0.00***	6.44	
Total Score	Baseline	46.62	6.76	44.23	12.04	−2.39	0.35	–	
After 6th week	22.14	5.74	33.80	10.58	11.65	0.00***	8.64	
Notes.

AKE active knee extension test

VAS visual analogue scale

WOMAC Western Ontario and McMaster Universities Arthritis Index

Differences within groups were analyzed by independent sample t-test.

*** p < 0.001.

Discussion

This study was conducted to determine the effectiveness of the Instrument-Assisted Soft Tissue Mobilization (IASTM) technique and PNF stretching in improving hamstring flexibility, pain, and health status in knee osteoarthritis. In the present study within group analysis showed that participants in both groups had significant improvement in pain, hamstring flexibility, and health status throughout the treatment duration. Between group analysis showed that IASTM technique showed more significant improvement in hamstring flexibility, pain, and health status on WOMAC scores (pain, stiffness and physical functions) as compared to PNF stretching.

A study comparing the effects of static stretching with PNF stretching in patients with knee osteoarthritis showed significant improvement in pain and hamstring flexibility in the PNF group (Meena, Shanthi & Madhavi, 2016). These results are consistent with the results of our study in which PNF stretching had effects in improving hamstring flexibility and pain score. Its neurophysiological mechanism, notably the activation of the Golgi tendon organ and the onset of anticipatory relaxation in the involved muscle, is responsible for PNF stretching’s efficiency in enhancing hamstring length. After numerous sessions, the sarcomeres and musculotendinous unit were significantly improved in the hamstring by the hold-relax approach utilized in PNF stretching (Meena, Shanthi & Madhavi, 2016; Reiner et al., 2021).

Within-group analysis showed that PNF stretching had significant results in reducing pain and improving health status. A study had similar results showing that PNF stretching significantly reduced pain intensity and improved functional performance in patients with knee osteoarthritis and chronic non-specific low back pain (Dharmendrasinh, 2020; Gul, Khan & Rahman, 2015). The increased blood flow to the muscles could be the cause of the pain reduction. Substance P, a neurotransmitter involved in the transmission of pain signals, may be removed as a result. When substance P is taken away from the injured location, it may have an analgesic effect that lessens pain perception. This mechanism shows that the PNF-hold relax approach may have a pain-relieving effect by modulating substance P levels and blood flow in addition to improving muscle relaxation (Meena, Shanthi & Madhavi, 2016).

An RCT showed that IASTM reduced pain and reduced functional disability in the patellofemoral pain syndrome (Dharmendrasinh, 2020). Another study concluded that IASTM has significant improvement in reducing pain, functional disability, and improving hamstring flexibility in non-specific low back pain (Heggannavar & Metgud, 2019). The IASTM helps to reduce pain by functioning as a neuromodulator, potentially reducing the activation of both big and tiny fiber neurons and producing an analgesic effect. Additionally, by mobilizing the fascia IASTM is believed to dissolve structural adhesions within the fascia. This procedure can improve hamstring flexibility and speed up tissue repair (Karmali, Walizada & Stuber, 2019; Heggannavar & Metgud, 2019; Gunn et al., 2019). These processes collectively imply that IASTM might have analgesic effects in addition to beneficial effects on tissue repair and flexibility.

The present study showed that IASTM had significantly improved hamstring flexibility compared to PNF stretching. This difference could be attributed to IASTM’s ability to directly target specific areas of tightness or adhesions, stimulate fibroblastic activity, induce tissue reattachment, and lead to greater improvements in hamstring flexibility (Kim, Sung & Lee, 2017). The stainless-steel instrument used in IASTM provides targeted and deeper penetration, mobilizes soft tissue more effectively, and promotes restoration of the injured area through hyperemia and collagen tissue repairment (Simatou et al., 2020). The present study’s results align with a previous study showing that IASTM reduces functional disability compared to conventional therapy (Ganesh, Ayman & Allen, 2017). In this study, IASTM showed greater improvement in overall health status assessed by the WOMAC index compared to PNF stretching. The use of IASTM identifies adhesions, generates an inflammatory process, increases muscular blood supply, reduces tissue viscosity, and promotes collagen tissue repairment (Dharmendrasinh, 2020; Lambert et al., 2017). Restoring hamstring flexibility has been shown to reduce pain and stiffness in knee osteoarthritis and improve quality of life, as the hamstring is a major muscle in the knee joint (Fokmare Jr & Phansopkar, 2022; Hafez et al., 2013).

The limitations of this study were that it was conducted in only one clinical setting. Another limitation that may affect the prognosis of the results, is the inclusion of both unilateral and bilateral knee osteoarthritis including different grades of knee osteoarthritis.

Conclusion

This study indicated those with knee osteoarthritis benefited from both the IASTM technique and PNF stretching, resulting in increased hamstring flexibility, decreased pain, and enhanced general health. The IASTM technique, however, showed potential benefits over PNF stretching in terms of flexibility, pain relief, and general health enhancement. These benefits could be attributed to the IASTM technique’s improved hamstring flexibility. Multicentered future studies are recommended on the bases of BMI, gender differences as well as retention effects of the intervention.

Supplemental Information

Supplemental Information 1 CONSORT Checklist

Click here for additional data file.

Supplemental Information 2 Trial Protocol

Click here for additional data file.

Supplemental Information 3 Raw data

SPSS or PSPP is required to view the dataset

Click here for additional data file.

Thanks to the participants of this study for sharing their personal experiences with pain.

Additional Information and Declarations

Competing Interests

Author Contributions

Human Ethics

Clinical Trial Ethics

Data Availability

Clinical Trial Registration

The authors declare there are no competing interests.

Narmeen Anjum conceived and designed the experiments, performed the experiments, analyzed the data, authored or reviewed drafts of the article, and approved the final draft.

Raheela Kanwal Sheikh conceived and designed the experiments, analyzed the data, authored or reviewed drafts of the article, and approved the final draft.

Aadil Omer conceived and designed the experiments, prepared figures and/or tables, authored or reviewed drafts of the article, and approved the final draft.

Kinza Anwar performed the experiments, analyzed the data, prepared figures and/or tables, authored or reviewed drafts of the article, and approved the final draft.

Muhammad Manan Haider Khan performed the experiments, authored or reviewed drafts of the article, and approved the final draft.

Anam Aftab analyzed the data, prepared figures and/or tables, authored or reviewed drafts of the article, and approved the final draft.

Waqar Ahmed Awan conceived and designed the experiments, performed the experiments, analyzed the data, prepared figures and/or tables, authored or reviewed drafts of the article, and approved the final draft.

The following information was supplied relating to ethical approvals (i.e., approving body and any reference numbers):

The Research and ethical Committee (REC) of Faculty of Rehabilitation and Allied Health Sciences of RIphah International University approval to carry out the study (with Ref# Riphah/RCRS/REC-01055).

The following information was supplied relating to ethical approvals (i.e., approving body and any reference numbers):

The Research and ethical Committee (REC) of Faculty of Rehabilitation and Allied Health Sciences of RIphah International University approval to carry out the study (with Ref# Riphah/RCRS/REC-01055).

The following information was supplied regarding data availability:

The raw data is available in the Supplemental File.

The following information was supplied regarding Clinical Trial registration:

NCT05110326

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
