# Peer review of "Comparison of instrument-assisted soft tissue mobilization and proprioceptive neuromuscular stretching on hamstring flexibility in patients with knee osteoarthritis"

_PeerJ, doi:10.7717/peerj.16506_

## Round 0.1 · original submission · Major Revisions

Both reviewers have important questions regarding this manuscript. They have also made suggestions that will help greatly in improving it. Please ensure grammar and punctuation errors are resolved as well as acronyms including PNF, AKE and IT, for example.

Reviewer 1 ·

Basic reporting

The paper would benefit greatly from editing by a fluent English speaker, as there are a number of errors that need attention. Some of these are word omissions, so the editor should be familiar with the research area.

Experimental design

Appropriate randomization and allocation to groups for a clinical trial was carried out.

Validity of the findings

The findings are valid and credible.
However, some of the post-hoc analysis carried out here is unnecessary. Significant interactions are differences in differences. Because there are two groups and two occasions of data-gathering, the interaction term has only one degree of freedom, so the significant interaction F-ratios can be unambiguously interpreted without the need for post-hoc t-tests. From Figure 4, it seems that the Baseline to 6 weeks difference for the IASTM group was greater than the difference for the PNF group on several measures. So the appropriate conclusion, which is stated in lines 247- 249, is given by the ANOVA results.

Additional comments

A main component of this paper is the stated use of the Ergon technique, which, the authors say, is also known as the Graston technique, and is an Instrument-Assisted Soft Tissue Mobilization (IASTM) technique (l.102-103). However, a search indicates that these two techniques are not the same.
In their review of IASTM techniques, Cheatham, Lee, Cain & Baker (2016) do include the Graston ® Technique, stating that there is a Graston company that has instruments of specific design. Notably, whenever the word Graston appears in their article, it is accompanied by the ‘registered trademark’ ® symbol, meaning that this brand name is protected by the US Patent and Trademark Office. In the current paper, this symbol is used in the Nejo reference in line 406, although the journal information is missing here.
The grastontechnique.com website indicates that in the 1990s David Graston developed the Graston Technique®, which uses a set of stainless steel instruments to perform muscle mobilizations.
Cheatham, Lee, Cain & Baker (2016) state that the Graston® Technique contains a protocol for treatment that has components of examination, warm-up, IASTM treatment, stretching, strengthening and ice when there is inflammation. In short, saying that the Graston® Technique has been used implies that a specific protocol has been followed.
Importantly, Cheatham et al note that IASTM treatments vary in approach and design, but that all have the general premise of enhancing myofascial mobility. This suggests that ‘IASTM’ is an overall heading that covers both branded and unbranded techniques.
Similarly, the brand named Ergon® Technique is a registered trademark. It is also an IASTM treatment approach for musculoskeletal rehabilitation, using specially-designed Ergon ® tools. The ergontechnique.com website indicates that K. Fousekis and K. Mylonas studied older IASTM techniques and developed their Ergon® Technique for IASTM in 2014. At their clinic in Greece, they train therapists in the use of the Ergon Technique. Clinicians using the technique apply Ergon® Cream and then select the appropriate size and shape of Ergon® tool to apply Ergon strokes of a specific angle of implementation and duration. Again, it would appear that saying that the Ergon ® Technique has been used implies that a very specific protocol has been followed by an Ergon-trained clinician, with specific instruments.
The solution for avoiding the Trademark and Copyright and any legal issues associated with registered brand names would seem to be to remove all reference to Ergon® from the paper and the title, simply saying that the treatment applied was IASTM, and specifying the contoured stainless steel instrument used, possibly with a photograph. In line 268 the IASTM tool is specified for another study, but not specified in the Methods here. Information about the lubricant (Vaseline), angle of implementation (45 degrees), and number of strokes is already given in the Methods section (l. 160-166). All of the description given here under the heading ‘Ergon Technique’ just refers to IASTM, with nothing specific to the Ergon® Technique, so removing all reference to Ergon will not be a problem.
Thus the title of the paper would become…. ‘Comparison of Instrument-Assisted Soft Tissue Mobilization and Proprioceptive Neuromuscular Facilitation techniques for improving hamstring flexibility in patients with knee osteoarthritis’.
.

Reviewer 2 ·

Basic reporting

General comments
In the study entitled “Comparison of Ergon technique versus proprioceptive neuromuscular facilitation stretching on hamstring flexibility in patients with knee osteoarthritis” aims to compare the effect of two different interventions on pain, hamstring length and health status in KOA. However, the paper lacks of methodological rigour. And, throughout the full manuscripts, I miss a clear rationale for conducting this study. The main reasons are present below:

Experimental design

Specific comments (**line numbers are based on the PDF copy)
Abstract
The demographic data are not the main indicators and can be briefly mentioned in this section. However, the most important part of the intervention approach has been ignored, such as the duration, and frequency.

According to the abstract, the author does not understand the main content of the study.

Introduction
Similar with the introduction section in abstract, this introduction does not provide a clear rationale to why this study should be performed. As mentioned by authors, hamstring muscle tightens to increase the patello-femoral compressive force, whether the function of the posterior femoral muscle group can be considered more related to the patellofemoral joint? At present, it is widely believed that quadriceps muscle function plays a more important role in the occurrence and development of KOA. Therefore, the authors are recommended to provide more evidence to demonstrate the significance of this study.

Did you have a hypothesis or research question? Please add the end of the introduction section.

Methods
Line 131, are only the patients with hamstring tightness more than 20°included? Why?
And what’s the AKE test? In abstract section, it refers to measure the muscle length.

How did you decide on the number of cases? Are these individuals sufficient for this study?

Treatment Group A and B should be clarified in the abstract section.

How were the two intervention program determined? Why did participants in Group A conduct Ergon technique, while Group B performed both PNF stretching and strengthening exercises? And how about the duration and repetition?

It’s suggested that a blind control group should be included in this study.

Is there a difference in gender, and the hamstring tightness among groups?

Validity of the findings

Results
It is recommended to keep two decimal places.

Can the results of the questionnaire be used as graded data by ANOVA?

Discussion
The authors should pay more attention on the effect of Ergon technique on KOA patients. The main reason is that the author did not explain the relationship between the flexibility of hamstring and KOA in the article.

Clarify why there is a discrepancy between studies? Were participants different? Were interventions or outcome measures different? As is, the discussion of past literature is rather shallow.

The authors seem to have confused muscle length with flexibility.

---

## Round 0.2 · Minor Revisions

The reviewers have raised some important points for further edits. Please address these.

Reviewer 1 ·

Basic reporting

Clear academic research writing

Experimental design

No comment

Validity of the findings

No comment

Additional comments

I would like to thank the authors for responding so thoroughly and carefully to my previous revision requests. The following are some minor remaining edits that are needed.

SPECIFIC EDITS
l.85 Modern types of these
l.99 It may help
l.136 The sealed envelope method
l.145 A hot pack
l. 164 stretch his or her hips
l.170 They held the stretch
l.199 comparisons, while
l.217 group-wise
l.247 are consistent
l.254 Within-group
l.299 are recommended on the basis of gender differences

Reviewer 2 ·

Basic reporting

The manuscript in its current version has been considerably improved. The material and method section has a new order that favors the reading and understanding of the research carried out. However, there are a couple of issues should be modified, such as statistical methods and control group settings.

Experimental design

Abstract
The essential elements of the intervention content were still not added, such as the duration, and frequency, which may not add much to the word count.

Line 35, “IASTM” refers to instrument-assisted soft tissue mobilization?

The Results of abstract section, line 42 and line 46, as confirmation of methods, testing methods may not need to be repeated.

The conclusion section should be modified to enhance the clinical significance.

Introduction
Line 61, sex, or female gender? Please check the description.

Moreover, the consistency of professional terms should be checked, for example, KOA and knee OA.

As mentioned in line 61-67, several factors has been demonstrated to be associated with KOA, whether the subjects in this study were compared and analyzed for related factors between groups?

Methods
Line 138, how did the authors determine the effect size of 0.24?

A schematic diagram may help to tell the differences between the two intervention groups in this study, including hot packs, strength training, and so on.

Validity of the findings

The WOMAC is a five-point scale, and continuous data are not available. Therefor, a statistical expert is recommended for futher confirmation.

Additional comments

The authors are advised to review the whole to avoid inconsistencies in the description, such as "Ergon technique" in the conclusion section.

---

## Round 0.3 · accepted · Accept

Please make one change to line 155. Rather than saying Baseline was assessed at the baseline..." say " Baseline assessments were made prior to beginning the intervention..."